# Unlocking the Nexus between Leaf-Level Water Use Efficiency and Root Traits Together with Gas Exchange Measurements in Rice (*Oryza sativa* L.)

**DOI:** 10.3390/plants11091270

**Published:** 2022-05-09

**Authors:** Ramasamy Gobu, Goutam Kumar Dash, Jai Prakash Lal, Padmini Swain, Anumalla Mahender, Annamalai Anandan, Jauhar Ali

**Affiliations:** 1Crop Improvement Division, Indian Council of Agricultural Research (ICAR)-National Rice Research Institute (NRRI), Cuttack 753006, Odisha, India; gobu.agri@gmail.com (R.G.); gkdash.bot@gmail.com (G.K.D.); 2Division of Crop Improvement and Biotechnology, Indian Council of Agricultural Research (ICAR)-Indian Institute of Spices Research (IISR), Kozhikode 673012, Kerala, India; 3Crop Physiology and Biochemistry Division, Indian Council of Agricultural Research (ICAR)-National Rice Research Institute (NRRI), Cuttack 753006, Odisha, India; pswaincrri@gmail.com; 4Department of Genetics and Plant Breeding, Institute of Agricultural Sciences, Banaras Hindu University, Varanasi 221005, Uttar Pradesh, India; jplalbhu@gmail.com; 5Rice Breeding Innovation Platform, International Rice Research Institute (IRRI), Los Baños 4031, Philippines; m.anumalla@irri.org; 6Indian Council of Agricultural Research (ICAR)-Indian Institute of Seed Science (IISS), Bangalore 560065, India

**Keywords:** rice, water use efficiency, deep rooting, drought stress, biomass, root traits

## Abstract

Drought stress severely affects plant growth and development, causing significant yield loss in rice. This study demonstrates the relevance of water use efficiency with deeper rooting along with other root traits and gas exchange parameters. Forty-nine rice genotypes were evaluated in the basket method to examine leaf-level water use efficiency (WUEi) variation and its relation to root traits. Significant variation in WUEi was observed (from 2.29 to 7.39 µmol CO_2_ mmol^−1^ H_2_O) under drought stress. Regression analysis revealed that high WUEi was associated with higher biomass accumulation, low transpiration rate, and deep rooting ratio. The ratio of deep rooting was also associated with low internal CO_2_ concentration. The association of deep rooting with lower root number and root dry weight suggests that an ideal drought-tolerant genotype with higher water use efficiency should have deeper rooting (>30% RDR) with moderate root number and root dry weight to be sustained under drought for a longer period. The study also revealed that, under drought stress conditions, landraces are more water-use efficient with superior root traits than improved genotypes.

## 1. Introduction

The impact of climate change and rapid human population growth has imposed a strong challenge for sustainable agriculture [1,2]. Climate change, increasing water shortages, more frequent drought, and high temperatures cause significant loss in grain yield [3,4]. Agriculture consumes approximately 70% of freshwater annually [5]. The major crop rice requires a large amount of water [6] and is highly susceptible to drought stress. It is estimated that approximately 2500 L of water are required to produce 1 kg of rice [7]. Rice production is particularly affected in rainfed ecosystems where drought is not uncommon due to erratic rainfall caused by climate change [4,8]. However, different strategies such as the use of sprinklers and drip irrigation to minimize excessive water use [9,10], antitranspirants [11,12], and variety development through conventional breeding [13] were explored to improve water use efficiency (WUE). The most plausible way is the genetic approach, which would be easy to adapt to minimize water use [14]. Therefore, identifying genotypes with an inherent ability to use water efficiently is a promising approach.

Plants exchange gases and water vapor through stomata, thereby regulating canopy temperature by controlling transpiration. When water is limited, transpiration is restricted by the closure of stomata, which affects carbon assimilation and increases canopy temperature [15,16]. The transpiration rate is dependent on the vapor pressure deficit (VPD). Genotypes tend to transpire more when the VPD is higher. Drought-tolerant genotypes can restrict their transpiration rate after a threshold VPD value by the partial closing of stomata, but susceptible genotypes continue to transpire even at a high VPD, resulting in severe water loss and wilting [17]. Severe water loss leads to a higher canopy temperature in susceptible genotypes than in tolerant ones under drought. In addition to severe water loss, the complete closure of stomata in susceptible genotypes inhibits carbon assimilation. Therefore, stomata play an important role in regulating water use efficiency in rice [14].

In the post-Green Revolution period, more emphasis was given to increasing grain yield. With the gradual rise in problems caused by global climate change, loss in grain yield was observed due to adverse environmental factors, particularly drought, for which the genotypes released were not adaptable to drought stress conditions. To mitigate this, plant breeders began to identify genotypes with an inherent capacity to tolerate drought, along with the putative linked QTLs. Since drought tolerance is a complex trait, no candidate genes have been identified to date for drought tolerance; rather, QTLs have been identified for traits associated with drought tolerance. In recent decades, the characterization of root architecture traits has become the target for breeders to enhance tolerance of drought and nutrient deficiency [18]. Several root-attributed QTLs have been identified in breeding lines and diverse rice germplasm [19,20,21,22,23]. These QTLs explained the role of root traits in drought tolerance. For the past two decades, efforts have been made to develop drought-tolerant rice varieties through the characterization and manipulation of root traits using breeding approaches [23,24,25,26,27,28,29,30]. The QTL Dro1 is one of them. It controls root growth angle and is responsible for deeper rooting in rice [31]. Dro1 is reported to minimize water loss and maintain a lower canopy temperature in rice by maintaining the water status of the plant under drought. However, the effect of rooting patterns on water use efficiency is a little explored area. Dharmappa et al. [13] reported a 50% increase in root dry weight and a 23.5% increase in root length, resulting in a significant increase in WUE. Similarly, Zhou et al. [32] reported that deeper root distribution resulted in increased WUE in super rice. However, the relevance of deeper rooting to water use efficiency is debatable. Our study demonstrates the relevance of WUE to deeper rooting along with other root traits and gas exchange parameters.

## 2. Results

### 2.1. Genotype Performance and Significance of ANOVA for Gas Exchange Measurements and Root Traits

A total of 49 rice genotypes that included 20 landraces and 29 improved varieties were evaluated for their deeper rooting using the basket method. Root samplings were taken from a depth of 50 cm from the soil surface for both well-watered and drought stress conditions. Highly significant differences (*p* < 0.0001) in various trait values such as photosynthesis rate, transpiration rate, leaf-level water use efficiency, internal CO_2_ concentration, shoot dry weight, root dry weight, ratio of deep rooting, root number, and root diameter were observed between the two treatments. Significant differences were also observed between the improved genotypes and landraces in all traits under well-watered (WW) and drought stress (DS) conditions (Table 1). The genotypes were exposed to drought stress for 20 days, during which the soil moisture content dropped to 11.20% at 30 cm and 16.39% at 45 cm of soil depth. As a result, the soil moisture tension dropped to −62.9 kPa and −51.9 kPa at 30 cm and 45 cm of soil depth, respectively, which was sufficient to cause moderate to severe drought stress that can affect plant growth significantly (Figure 1A,B). Drought stress diminished the average shoot biomass up to 51.6%. The decrease in shoot dry weight (SDW) was 56.1% in improved genotypes, whereas landraces registered a decrease of 45.7% under drought stress. Among all the genotypes, AC 35717 (12.1%), AC 36762 (12.6%), Moroberekan (19.8%), AC 36735 (23.8%), AC 36734 (26.9%), AC 35729 (28.1%), and AC 36738 (30%) had a minimal decrease in SDW and can thus be considered as drought tolerant (Table 2).

### 2.2. Variation in Instantaneous WUE (WUEi) and Biomass among Rice Accessions under Varying Moisture Levels

The WUEi values varied significantly (*p* < 0.0001) among genotypes (Table 1) and a significant increase in WUEi (*p* < 0.0001) was observed under drought conditions. During the entire investigation period, WUEi varied from 2.02 to 4.54 µmol CO_2_ mmol^−1^ H_2_O under well-watered conditions and from 2.29 to 7.39 µmol CO_2_ mmol^−1^ H_2_O under drought stress conditions. No significant variation in WUEi was observed between the improved lines (2.95 µmol CO_2_ mmol^−1^ H_2_O) and the landraces (3.01 µmol CO_2_ mmol^−1^ H_2_O) under WW conditions. However, under drought conditions, the landraces (4.70 µmol CO_2_ mmol^−1^ H_2_O) showed significantly higher WUEi than the improved genotypes (3.54 µmol CO_2_ mmol^−1^ H_2_O). Among the improved genotypes, Ranbir Basmati 2 alone registered >5 µmol CO_2_ mmol^−1^ H_2_O, while five landraces (AC 36762, AC 36734, AC 35717, AC 36735, and AC 36722) had higher WUEi under DS, from 5.42 to 7.39 µmol CO_2_ mmol^−1^ H_2_O (Table 3).

Similarly, a significant variation was observed in shoot biomass among the genotypes. Biomass accumulation varied significantly between the improved genotypes and the landraces. The improved genotypes had a significantly higher shoot biomass, ranging from 28.05 g to 58.68 g with an average value of 41.92 g under WW conditions, whereas the landraces had a shoot biomass ranging from 14.23 g to 55.92 g, with an average value of 37.96 g. In contrast to WW conditions, landraces had a much higher value for shoot biomass under DS conditions, ranging from 5.16 g to 42.31 g, with an average value of 21.06 g compared with the average value of 18.38 g in improved genotypes. Among the landraces, AC 35717 (42.31 g) and AC 36762 (38.8 g) had the highest SDW. CR Dhan 207 (26.6 g) and Sattari (25.3 g) had the highest SDW among the improved genotypes.

### 2.3. Differences in Root Traits among Genotypes under Varying Moisture Levels

Variation was much wider for the ratio of deep rooting (RDR) in landraces (5.36–40.00%) than in improved genotypes (5.95–24.00%). Six landraces (AC 35717, AC 36762, AC 36735, AC 35729, Moroberekan, and AC 36738) exhibited >30% RDR, while Ranbir Basmati 2 alone had a higher RDR of 24% in the category of improved genotypes (Table 2). Similarly, significant variation was observed for root dry weight (RDW) (*p* < 0.003) and total root number per plant (RN) (*p* < 0.041) between treatments, as well as between landraces and improved genotypes. However, a significant decrease in RDW was observed irrespective of groups under DS. The average decrease in RDW was lower in landraces (19%) than in improved genotypes (21%). On the contrary, the RN observed in improved genotypes (74) was much higher than that of landraces (61) under drought (Table 2).

### 2.4. Correlation between Different Gas Exchange Traits and Root Traits

The correlation among the different studied traits is presented in the form of a network to visualize the interrelationship among the traits. Under WW conditions, RN, RDW, and RDR were correlated with each other. Similarly, WUEi, Ci, and Tr were intercorrelated. SDW showed a relatively weak correlation with other traits. However, An had a strong positive correlation with WUEi and a weaker correlation with Ci, SDW, and RN (Figure 2A–C).

Under DS conditions, all the traits (WUEi, Ci, An, RDW, RDR, RN, and SDW) were strongly intercorrelated with each other except Tr. Tr had a comparatively weak correlation with the other traits (Figure 2D–F). RDR had a strong positive correlation with WUEi, SDW, and An. However, it had a strong negative correlation with RDW, Ci, and RN.

### 2.5. Instantaneous WUE (WUEi) Is Correlated with the Ratio of Deep Rooting and Biomass Accumulation

Instantaneous WUE is the ratio of photosynthetic rate to transpiration rate. High WUEi results from a higher photosynthetic rate with the same transpiration rate, the same photosynthetic rate with a low transpiration rate, or both. To analyze the factors that contribute more to WUEi, both transpiration rate and photosynthetic rate were regressed against WUEi (Figure 3). WUEi was strongly associated with photosynthesis (r = 0.701 **) and transpiration rate (r = −0.484 **) under drought conditions. This suggests that higher photosynthetic rates and lower transpiration rates contribute to higher WUEi. A higher photosynthetic rate is considered an indicator of high biomass accumulation that leads to higher WUEi, which is evident from the strong positive correlation of WUEi with shoot biomass accumulation (r = 0.634 **). At the same time, RDR showed a strong positive correlation with WUEi (r = 0.708 **) and shoot biomass (r = 0.788 **), thus indicating that deep-rooted genotypes efficiently use water to produce higher biomass under drought (Figure 3A,B). Further, the genotypes with high RDR had a lower decrease in shoot dry weight, which is an indicator of drought tolerance. For instance, traditional landraces AC 36738, AC 35729, AC 35717, AC 36762, AC 36735, and Moroberekan exhibited >30% RDR with a minimum reduction in SDW, ranging from 12.1% to 28.1%.

### 2.6. RDR Is Associated with Low Internal CO_2_ Concentration, Root Dry Weight, and Total Number of Roots

The ratio of deep rooting significantly affected the internal CO_2_ concentration (Table 3). RDR showed a strong negative association with Ci (r = −0.75 **) (Figure 4). The genotypes with more than 30% RDR had comparatively lower Ci (214−264 μmol CO_2_ mol^−1^ H_2_O) under drought. Similarly, RDW (r = −0.398 *) and RN (r = −0.386 *) revealed a negative association with RDR. The genotypes with greater RDR had relatively lower RDW and RN (Figure 5A,B).

### 2.7. PCA with Root Traits Grouped the Genotypes into Distinct Categories Highlighting the Importance of RDR

To study the variance existing with these three root-related traits (RDR, RDW, and RN), the genotypes were plotted in a biplot that explained 80.11% of the total variance, according to the first two factors (F1 explained 41.59% and F2 explained 38.52% of the total variance). The biplot clearly separated the 49 genotypes into three groups. The first group included 25 genotypes showing high values for RDW and RN but low RDR. The second group included six genotypes with a high RDR and intermediate values for RDW and RN (AC 35717, AC 36762, AC 36735, AC 35729, AC 36738, and Moroberekan). The third group included 18 genotypes with low values for RDR, RDW, and RN (Figure 6).

### 2.8. Changes in Root Traits in Improved Genotypes from 1987 to 2016

To examine the effect of the selection pressure of grain yield on root traits, changes in RDR, the total number of roots in the crown region, and a root diameter within 10 cm of soil depth from the ground were analyzed in 24 genotypes released from 1987 to 2016. As the year of release of IR119, Phaninder, VL Dhan 08, IR36; and Ranbir Basumati (Basmati) 2 was unknown, these varieties were not included in this study. A gradual increase in RDR was observed from 8.5% to 13.0% within the past three decades, and a similar trend was observed for a root diameter present within 10 cm of soil depth from the ground (data not shown). The root diameter increased from 0.18 cm to 0.31 cm. In contrast, the number of roots in the crown region declined from 86 to 66 (Figure 7A,B). As shown in Figure 7A, there was no change in RDR from 1987 to 2006 but there was a considerable increase in RDR in the varieties released after 2009 (Figure 7B). Similarly, there was a sharper increase in root diameter from 2009 to 2016 than from 1987 to 2006.

## 3. Discussion

In the scenario of current climate change and in order to improve WUE in rice, deeper rooting is strongly required. In addition to water absorption, deeper rooting helps acquire nutrients from deeper soil layers. Nutrients such as nitrates are highly mobile and leach deep into the soil with percolating water, and these nutrients can be absorbed by deeper roots [33]. Maintaining a high rate of photosynthesis with a high biomass under decreased water is challenging in rice [34,35]. Efficient water use by the crop is accomplished through stomatal regulation and a reduction in stomatal density [14]. A reduction in stomatal density and transpiration check through stomatal regulation might decrease unnecessary water loss [17,36] but result in high canopy temperature under WW conditions, compared to those plants showing high stomatal density without any stomatal regulation [14,37]. Increased WUE might result from a high photosynthetic rate at the same transpiration rate or the same rate of photosynthesis from a low transpiration rate. The combination of both high photosynthetic rate and low transpiration rate can contribute to higher WUE in the active tillering stage. However, the significant contribution between these two factors toward high WUE is still debatable. During evolution, plants have maximized WUE through sustained photosynthetic rate and low transpiration rate [12]. We suggest that, although the contribution of both factors is significant, the rate of photosynthesis (*r* = 0.701 **) contributes more to WUEi than the transpiration rate (*r* = −0.484 **). Although this conservative water strategy is useful for survival under drought conditions, a decrease in transpiration rate restricts CO_2_ influx, resulting in decreased growth [13]. On the other hand, efficient water use is relevant with a higher growth rate [38]. In our opinion, for efficient water use, plants have to extract more water from the soil and diminish unnecessary water loss. Tolerant genotypes with restricted transpiration above a threshold VPD can maintain their photosynthetic rate without affecting biomass production, despite having partial stomatal closure under well-watered conditions. Under drought stress conditions, the tolerant genotypes with high RDR extract water from deeper soil layers and thus maintain the water status of the plant and restrict transpiration to decrease unnecessary water loss, thereby enabling them to photosynthesize for a longer period. However, the susceptible genotypes lack this mechanism. Our study found a strong correlation of WUEi with RDR under DS. Deeper rooting helps to siphon water from the deeper soil layers, thus maintaining turgidity in plants during drought and favoring comparatively higher photosynthesis. Usually, deeper root growth lowers water use efficiency, but in our study higher RDR favored higher WUE because of higher photosynthetic rates. The restriction in transpiration through stomata is independent of water extraction by roots, evident from the lack of correlation between transpiration rate and RDR (r = 0.074). Therefore, water use efficiency is a coordinated effort performed by both the root and stomata.

The contribution of deeper rooting toward WUEi is also supported by the strong negative correlation between RDR and Ci. Low Ci is an indicator of high carboxylation efficiency. The genotypes registering high RDR had low Ci under drought, thus explaining higher carboxylation efficiency and WUEi. Under drought conditions, the rapid closure of stomata occurs, which leads to a decrease in Ci, suggested by some researchers [13]. According to this statement, Ci is supposed to be low in all genotypes because of the drought effect. However, stomata remain open in rolled leaves and leaf rolling is not directly related to drought response [39], which explained the variability in Ci in the studied genotypes under drought. Further, high RDR is expected to lower WUEi because of more open stomata and increase Ci; however, negative correlation of Ci with RDR and WUEi exists as WUEi is driven by a high photosynthetic rate.

Furthermore, RDR was negatively associated with RDW (r = −0.398*) and RN (r = −0.386*). In our study, the genotypes with high RDR were not the highest scorers in RDW and RN. This is observed clearly in Figure 5. The group that had high RDR exhibited intermediate values for RDW and RN. Contrary to this, genotypes that had high RDW and RN had low RDR. Many researchers reported that genotypes with high RDW are considered drought tolerant [24,40,41]. In contrast, deep rooting in a thinner root system has been observed to tolerate stronger drought conditions [42]. Our study found that genotypes with deeper rooting did not develop a thicker or denser root system at the cost of shoot growth. Rather, they meticulously diverted biomass toward the root to produce a moderately thick root with a lesser root number to continue shoot growth under drought stress. This is evident from the low rate of shoot biomass reduction in genotypes with high RDR. The RN recorded in improved genotypes was much higher than that of landraces under drought. This indicates that they have more surface roots developed for higher input (transplanted) conditions and are able to absorb more nutrients from the sub-surface soil profile [43]. Hence, they are not suitable for dry direct-seeded rice (rainfed upland), for which high RDR is necessary. The genotypes suitable for direct-seeded rice conditions should possess more than 30% RDR with surface rooting to absorb nutrients and to have grain yield on a par with that of transplanted rice.

Drought is a multigene-governed phenomenon for which no specific candidate gene has been identified. To study drought tolerance, researchers focus on particular traits such as stomatal density, osmotic regulation, root traits, etc. In past decades, the rice varieties developed had high grain yield with good grain quality. These varieties were targeted for irrigated areas and were thus highly susceptible to drought, causing substantial yield loss [44]. Owing to directional selection for grain yield, root traits were completely neglected in breeding programs, thus making improved varieties more vulnerable to drought stress [40]. Since the climate change scenario has worsened in the past two decades, breeders now emphasize root traits for developing drought-tolerant genotypes. The recent varieties developed from ICAR-NRRI, Cuttack, after 2014 for direct-seeded aerobic conditions such as CR Dhan 203, CR Dhan 206, and CR Dhan 207 have a high root biomass with fewer, thicker, and deeper roots (Figure 6). However, dense roots with high root numbers may rapidly deplete the soil water level at the early vegetative stage, causing terminal drought stress where water availability or rain occurs at early developmental stages. In contrast, an intermediate number of crown roots with deeper roots of thicker diameters may conserve water for the later developmental stage. This type of root system would be useful for direct-seeded rice systems and an intermediate number of crown roots would help to absorb the nutrients required for higher output in contrast to genotypes with a high RDR [45]. In our study, the landraces mostly have good and adaptive root traits compared with the improved genotypes because of our breeding program, which was oriented toward achieving higher grain yield. This led to a reduced root system compared with landraces and wild ancestors [46,47]. In addition, a higher metabolic cost (root respiration ~50% of the daily photosynthesis) for root growth and maintenance was involved [48]. The selection for high grain yield and harvest index has promoted genotypes with less biomass allocation toward the root [49]. However, a low-cost root system can be designed with fewer lateral roots and deep roots suitable for drought conditions [50]. Further, comparatively higher root growth would help to sequester more carbon from the atmosphere in this era of climate change [45].

## 4. Materials and Methods

### 4.1. Plant Materials

A total of 49 rice genotypes comprising 20 traditional landraces and 29 improved genotypes (Table 2) were collected from the gene bank of ICAR-National Rice Research Institute (NRRI), Cuttack, India. The experiment was conducted in a cemented tank at NRRI during the wet season of 2017. Each genotype was raised in a plastic basket repressed in the soil of cemented tanks in a randomized block design. The cemented tank was 1 × 4 × 2 m in size filled with sandy clay loam soil and had a penetration resistance of 11.12 to 1356.22 kPa with an average of 648.54 kPa (cone penetrometer; CP20, Rimik, Australia). The baskets were 10 cm in height. The baskets’ top and bottom diameters were 20 cm and 10 cm, respectively, with a mesh size of 3 mm. Four seeds of each genotype were sown per basket with three replications. Separate tanks were maintained for drought stress and well-watered treatments. Seven days after germination, a single plant was maintained in each basket. Thirty days after germination, drought stress was imposed by withdrawing irrigation. During the stress period, soil moisture content was measured using an ML3 Theta probe soil moisture sensor (Delta-T, Cambridge, UK) connected to a GP1 data logger. In addition, soil moisture tension was measured using a tensiometer (Soilspec, Australia) to monitor drought stress intensity. The measurements were taken at soil depths of 30 cm and 45 cm. Stress was continued until the soil moisture content dropped below 14–16% at 45 cm of soil depth, while the WW tank was supplied daily and necessary plant protection measures were taken. Fertilizer was applied as per the recommended doses of N:P:K at 80:40:40 kg ha^−1^ in the form of urea, single superphosphate, and muriate of potash, respectively.

#### 4.1.1. Root Trait Measurements

After 20 days of stress (DAS) period (at tillering stage), one side of the cemented tank wall was broken and the soil around the baskets was removed carefully with the help of a jet pipe to minimize damage to the roots. The roots were excavated from a depth of 50 cm, amounting to 0.125 m^3^ of soil volume. The total number of roots and roots that emerged from the base of the baskets (i.e., 50° to 90° from the surface of the basket) were counted separately. The ratio of deep rooting was calculated as the ratio of the number of roots that emerged from the lower part of the baskets (the portion of the base of the baskets defined by an angle of 50° to 90° from the horizontal, taking the stem of the plant as the central axis) to the total number of roots emerged from the whole basket [31]. In addition, shoot dry weight (SDW) and other root-related traits such as root diameter (RD) (using a digital Vernier caliper), root dry weight (RDW), and total number of roots (RN) were measured. The samples were dried in a hot-air oven at 60 °C for 5 days to record RDW.

#### 4.1.2. Gas Exchange Measurements

Gas exchange was measured with the help of an infrared gas analyzer (IRGA) LI-6400 XT system (Li-Cor, Lincoln, NE, USA). This occurred under standard conditions at active tillering stage 10 days after stress imposition to avoid inhibition of photosynthesis due to leaf rolling. To avoid environmental fluctuations, 1500 μmol quanta m^−2^ s^−1^ of light was supplied through an RGB LED light source and a constant flow of 400 μmol CO_2_ mol^−1^ was supplied through a carbon dioxide cylinder attached to the system. The temperature of the leaf chamber was maintained at 25 °C with a flow rate of 500 μmol s^−1^. Measurements were made on the second fully expanded leaf after steady-state conditions were obtained (indicated by steady stomatal conductance and assimilation rate) from each replication. The photosynthesis rate (An, μmol CO_2_ m^−2^ s^−1^), transpiration rate (Tr, mmol H_2_O m^−2^ s^−1^), vapor pressure deficit (VPD), stomatal conductance (gs, mol H_2_O m^−2^ s^−1^), and internal CO_2_ concentration (Ci, μmol CO_2_ mol^−1^) were calculated by the manufacturer’s software. Instantaneous water use efficiency (WUEi) or leaf-level water use efficiency was calculated as the ratio between net photosynthetic rate (An) and transpiration rate (Tr).

### 4.2. Statistical Analysis

Significant differences between treatments and improved genotypes vs. landraces for the studied traits were estimated through a *t*-test using XLSTAT 2014 version (https://www.xlstat.com accessed on 6 July 2021) to assess the variability among the genotypes and the treatments. The interaction between traits and genotypes was established through principal component analysis and the scatter plots for correlation were constructed using XLSTAT 2014. Pearson phenotypic correlation and the network of morpho-physiological traits were assessed using the packages “*ggcorrplot*” and “*qgraph*” in R version 4.1.3 (R Core Team 2022).

## 5. Conclusions

Our study revealed sufficient variability in root traits and gas exchange parameters among genotypes in response to drought. The study found that deeper rooting helps to increase water use efficiency by lowering Ci under drought. The study also suggests that genotypes that are suitable for dry direct-seeded rice conditions should have a higher RDR and root diameter with a moderate number of roots. Previously, little attention was given to root traits in rice breeding programs. For the past decade, the rice varieties released have had deep roots with a comparatively low root number and high root diameter. These are more suitable to drought conditions in the current scenario of climate change.

## Figures and Tables

**Figure 1 plants-11-01270-f001:**
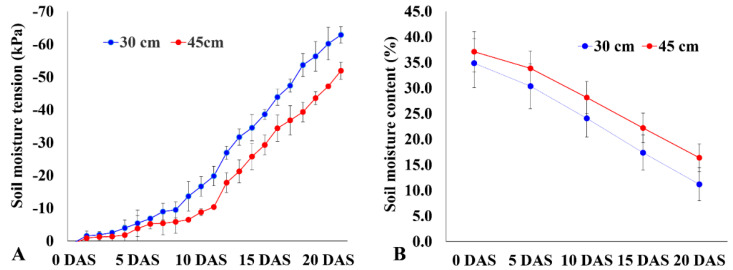
Soil moisture tension (**A**) and soil moisture content (**B**) measured by tensiometer and time-domain reflectometer, respectively, at soil depths of 30 and 45 cm. Values shown are mean ± SE, n = 3. DAS = days after stress.

**Figure 2 plants-11-01270-f002:**
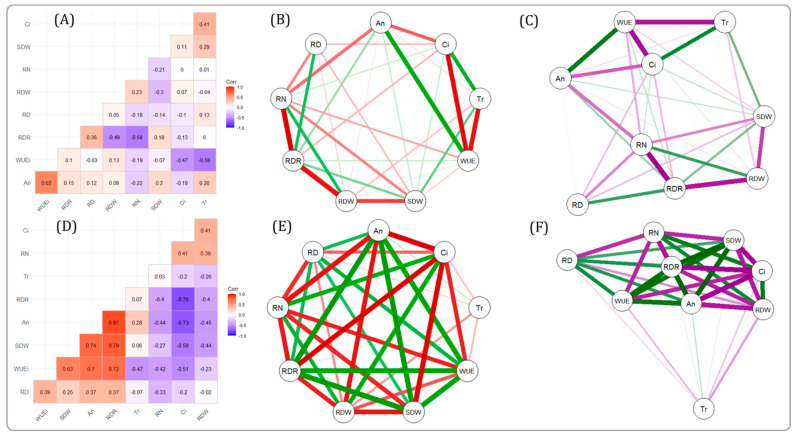
Pairwise Pearson phenotypic correlation and network of morpho-physiological traits. The correlation (**A**,**D**) and network (**B**,**C**,**F**) of the nine traits evaluated in rice plants under well-watered (WW) (**A**–**C**) and drought stress (DS) (**D**–**F**) conditions. The width of the lines represents the strength of the correlation and the color of the lines represents the nature of the correlation. The red and the purple lines represent negative correlation and the green line represents positive correlation.

**Figure 3 plants-11-01270-f003:**
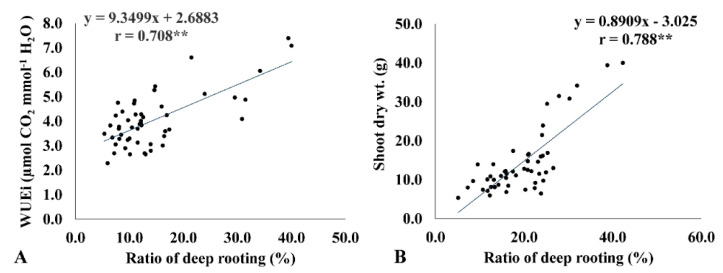
Linear regression relationship between the ratio of deep rooting (RDR) with instantaneous water use efficiency (WUEi) (**A**) and shoot dry weight (**B**) under drought conditions. ** <0.01.

**Figure 4 plants-11-01270-f004:**
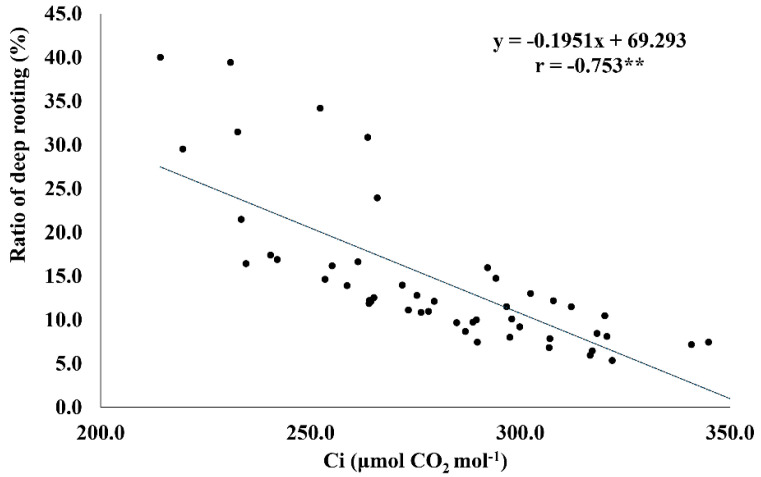
Linear regression relationship between the ratio of deep rooting (RDR) and internal CO_2_ concentration under drought conditions. ** <0.01.

**Figure 5 plants-11-01270-f005:**
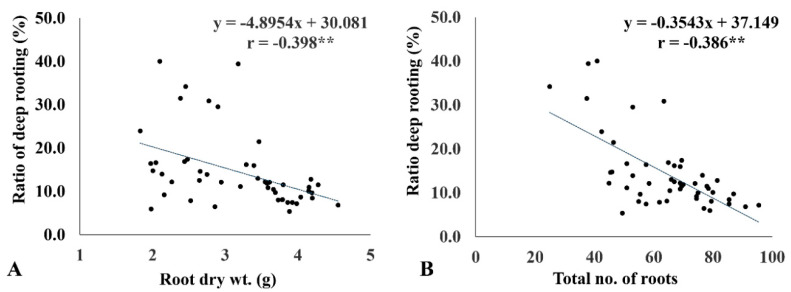
Linear regression relationship between the ratio of deep rooting (RDR) and root dry weight (RDW) (**A**) and the total number of roots (RN) (**B**) under drought conditions. ** <0.01.

**Figure 6 plants-11-01270-f006:**
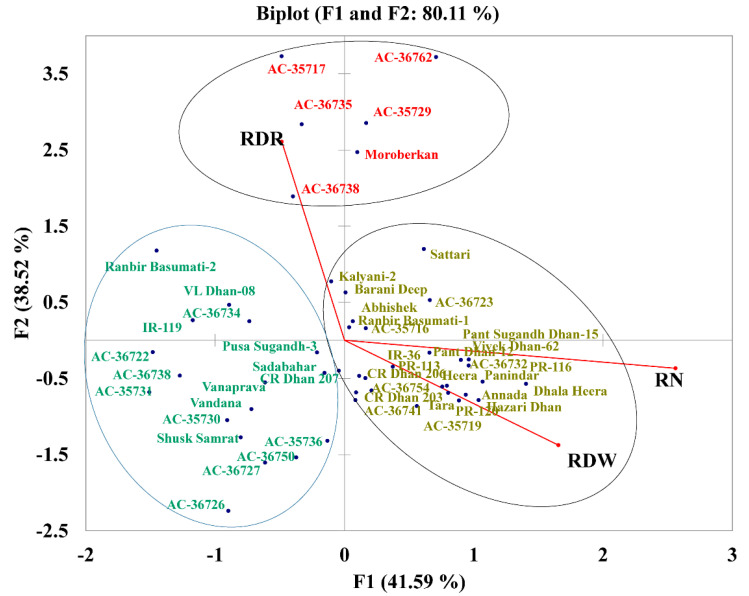
Factorial plans with two principal components representing the ratio of deep rooting (RDR), total number of roots (RN), and root dry weight (RDW) averaged over three replications of 49 rice genotypes under drought stress conditions. The variance explained by each dimension is shown as a percentage of the total variance.

**Figure 7 plants-11-01270-f007:**
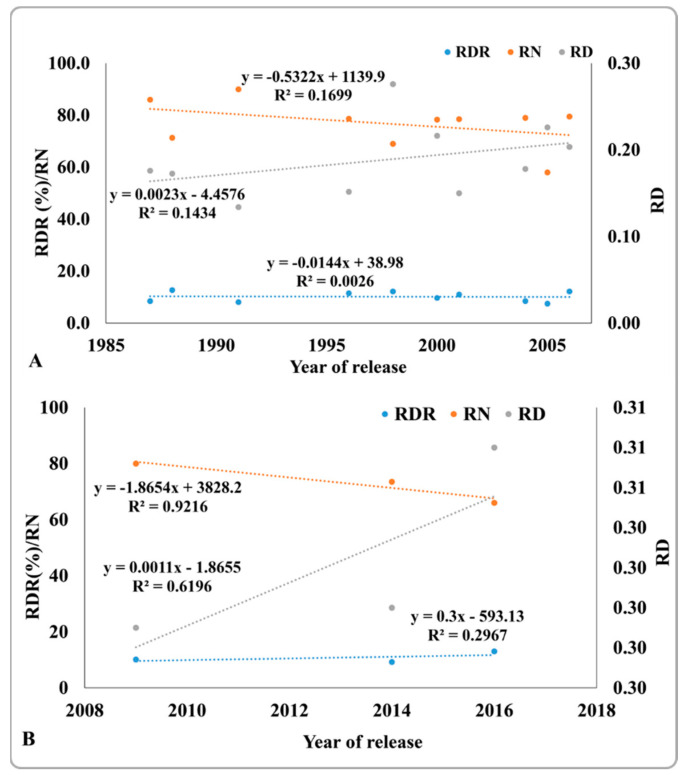
Ratio of deep rooting (RDR), total number of roots in the crown region (RN), and root diameter (RD) of varieties released from 1987 to 2006 (**A**) and from 2009 to 2016 (**B**). Values are the mean of three replicates per variety per year of release. Genotypes 1 to 24 from Table 2 and Table 3 were used here.

**Table 1 plants-11-01270-t001:** Overall significant difference level (*t*-test) between treatments and improved genotypes vs landraces for the studied traits.

Sl. No.	Traits	Treatments		Improved Genotypes vs. Landraces			
				WW		DS	
		df	*p* Value	df	*p* Value	df	*p* Value
1	Photosynthesis rate (An)	97	****	1	**	1	*
2	Transpiration rate (Tr)	97	****	1	**	1	*
3	Leaf-level water use efficiency (WUEi)	97	****	1	**	1	**
4	Internal CO_2_ concentration (Ci)	97	****	1	*	1	*
5	Shoot dry weight (SDW)	97	****	1	*	1	*
6	Root dry weight (RDW)	97	****	1	**	1	**
7	Ratio of deep rooting (RDR)	97	****	1	*	1	*
8	Root number (RN)	97	****	1	*	1	*
9	Root diameter (RD)	97	****	1	*	1	*

WW = well-watered; DS = drought stress; df = degrees of freedom; * <0.05; ** <0.01; **** <0.0001.

**Table 2 plants-11-01270-t002:** Variation in root and shoot parameters under well-watered and drought stress conditions in 49 rice genotypes.

Sl. No.	Genotypes	Type	SDW (g)			RDW (g)			RDR		RN		RD (mm)	
			WW	DS	PDR	WW	DS	PDR	WW	DS	WW	DS	WW	DS
1	CR Dhan 203	IP	40.6	23.9	41.2	3.89	2.86	26.5	10.3	6.5	84	77	0.425	0.325
2	CR Dhan 206	IP	42.0	25.0	40.6	4.28	3.58	16.5	13.9	11.9	82	70	0.368	0.275
3	CR Dhan 207	IP	43.7	26.6	39.1	3.89	3.45	11.3	16.4	13.0	79	66	0.425	0.308
4	Vandana	IP	45.0	22.5	50.1	3.12	2.53	19.1	12.4	7.9	81	62	0.345	0.186
5	Shusk Samrat	IP	44.4	20.3	54.2	3.54	2.93	17.4	13.2	7.5	73	58	0.369	0.226
6	Barani Deep	IP	37.3	13.1	64.9	1.93	2.13	−10.4	17.5	14.0	94	77	0.347	0.197
7	Pusa Sugandh-3	IP	53.3	22.6	57.6	3.72	2.16	41.9	14.3	9.2	78	75	0.345	0.166
8	PR-113	IP	37.3	15.9	57.5	4.99	3.55	28.9	12.4	12.2	112	69	0.452	0.276
9	Vivek Dhan-62	IP	49.2	20.0	59.3	4.23	4.18	1.2	13.5	12.8	96	82	0.389	0.134
10	Pant Dhan-12	IP	53.7	16.0	70.2	3.66	2.56	30.2	15.1	6.8	75	91	0.412	0.158
11	Annada	IP	48.4	16.5	66.0	5.25	4.2	20.0	15.4	8.5	93	86	0.384	0.176
12	Dhala Heera	IP	31.5	16.1	48.9	5.30	4.28	19.3	15.6	11.5	89	78	0.389	0.149
13	Heera	IP	35.6	13.4	62.3	4.57	3.79	17.2	15.4	8.1	99	90	0.375	0.134
14	Kalyani-2	IP	35.0	17.5	49.9	1.94	2.48	−27.8	10.3	17.4	126	70	0.384	0.223
15	Abhishek	IP	28.1	9.5	66.0	3.13	2.75	12.1	13.1	13.9	111	73	0.365	0.205
16	Pant Sugandh Dhan 15	IP	34.5	16.0	53.8	5.88	3.67	37.7	14.2	10.5	102	86	0.377	0.202
17	PR-116	IP	34.8	11.8	66.1	5.52	3.99	27.8	12.6	7.2	114	96	0.374	0.266
18	PR-120	IP	36.4	11.7	67.8	5.37	4.15	22.7	15.3	10.1	82	80	0.470	0.299
19	Ranbir Basmati-1	IP	54.2	24.2	55.3	4.85	3.29	32.2	12.1	16.2	94	67	0.425	0.148
20	Sattari	IP	48.8	25.3	48.1	2.33	2.44	−4.7	12.9	16.9	98	85	0.412	0.152
21	Tara	IP	30.9	14.3	53.8	5.44	4.04	25.7	10.2	8.7	127	85	0.421	0.149
22	Vanaprava	IP	45.4	15.7	65.5	3.74	2.95	21.3	11.9	12.1	100	59	0.423	0.146
23	Hazari Dhan	IP	38.1	14.8	61.2	6.19	4.16	32.9	15.8	11.0	72	79	0.418	0.181
24	Sadabahar	IP	33.6	12.3	63.4	4.25	1.98	53.4	15.1	6.0	95	79	0.471	0.175
25	IR119	IP	47.5	21.1	55.6	1.82	2.05	−12.4	10.4	16.6	108	51	0.402	0.184
26	Panindar	IP	47.2	24.3	48.4	3.92	3.69	5.9	13.6	9.7	82	87	0.389	0.256
27	VL Dhan-08	IP	58.7	20.9	64.4	3.30	1.98	40.2	14.5	16.4	96	58	0.398	0.258
28	IR36	IP	33.1	17.5	47.2	4.71	3.61	23.4	10.2	12.1	106	74	0.354	0.103
29	Ranbir Basmati-2	IP	47.7	24.3	49.1	1.48	1.83	−23.7	20.6	23.9	59	43	0.415	0.266
30	Moroberkan	LR	37.7	30.3	19.8	1.66	2.78	−67.2	27.6	30.9	78	64	0.489	0.430
31	AC 36722	LR	37.7	20.9	44.7	2.24	2.01	10.5	16.5	14.8	64	46	0.389	0.250
32	AC 36723	LR	49.1	20.8	57.6	3.77	2.65	29.8	14.3	12.6	98	87	0.471	0.390
33	AC 36732	LR	24.5	10.7	56.2	4.28	3.86	9.8	9.56	7.5	105	86	0.412	0.273
34	AC 36741	LR	23.5	12.4	47.2	5.77	3.59	37.8	12.7	10.9	87	69	0.463	0.422
35	AC 36726	LR	14.2	5.2	63.8	5.26	3.89	26.1	8.36	5.4	83	50	0.489	0.331
36	AC 36735	LR	42.0	32.0	23.8	2.24	2.46	−9.6	30.4	34.2	65	55	0.456	0.370
37	AC 36754	LR	55.9	23.4	58.1	5.04	3.80	24.6	13.9	11.5	86	70	0.425	0.356
38	AC 36750	LR	32.4	8.5	73.8	6.39	4.20	34.4	12.7	9.7	86	56	0.463	0.246
39	AC 36734	LR	32.9	24.1	26.9	3.88	3.47	10.7	19.4	21.5	63	47	0.398	0.356
40	AC 36727	LR	33.2	7.3	78.0	5.37	3.73	30.5	12.6	8.0	88	55	0.478	0.422
41	AC 36762	LR	44.4	38.8	12.6	2.84	3.18	−12.0	36.4	39.4	74	68	0.496	0.327
42	AC 35716	LR	43.8	23.8	45.6	4.77	3.40	28.8	18.6	16.0	83	69	0.462	0.306
43	AC 35717	LR	48.1	42.3	12.1	2.08	2.10	−0.96	37.6	40.0	63	51	0.460	0.347
44	AC 35719	LR	36.2	13.2	63.6	5.40	4.16	23.1	14.6	10.0	94	75	0.387	0.263
45	AC 35729	LR	38.8	27.9	28.1	2.01	2.39	−18.7	28.1	31.5	79	68	0.475	0.314
46	AC 35730	LR	35.5	18.2	48.7	5.22	3.21	38.5	16.4	11.1	76	51	0.384	0.260
47	AC 35731	LR	44.5	21.7	51.4	4.93	2.27	54.1	15.6	12.2	69	45	0.395	0.234
48	AC 36738	LR	34.4	25.2	26.9	1.53	2.90	−89.5	25.9	29.5	71	53	0.378	0.262
49	AC 35736	LR	38.4	12.6	67.2	4.25	3.79	10.8	13.8	8.1	96	65	0.375	0.254
	CD 5%		2.53	2.14		0.39	0.22		1.81	2.43	4.54	3.97	0.01	0.02

Improved varieties (IP); landraces (LR); percentage of reduction decrease (PRD); shoot dry wt. (SDW, g); root dry wt. (RDW, g); ratio of deep rooting (RDR); total number of roots at crown region (RN); root diameter (RD, mm); well-watered (WW); and drought stress (DS).

**Table 3 plants-11-01270-t003:** Variation in photosynthetic gas exchange traits under well-watered and drought stress conditions in 49 rice genotypes.

Sl. No.	Genotypes	Type	An		Ci		Tr		WUEi	
			WW	DS	WW	DS	WW	DS	WW	DS
1	CR Dhan 203	IP	22.8	11.3	244.3	317.2	7.86	2.94	2.90	3.83
2	CR Dhan 206	IP	24.5	11.2	172.2	264.0	7.24	2.87	3.39	3.88
3	CR Dhan 207	IP	24.7	10.9	252.0	302.5	8.25	4.12	2.99	2.65
4	Vandana	IP	26.6	12.6	134.5	307.1	7.22	2.66	3.68	4.75
5	Shusk Samrat	IP	22.1	10.7	155.1	289.8	6.75	3.51	3.27	3.05
6	Barani Deep	IP	25.2	11.3	232.3	271.9	9.23	4.03	2.73	2.80
7	Pusa Sugandh-3	IP	27.7	12.9	244.3	299.9	8.95	4.44	3.10	2.90
8	PR-113	IP	23.1	13.3	201.3	264.0	9.49	3.47	2.43	3.84
9	Vivek Dhan-62	IP	19.2	9.4	241.1	275.4	9.50	3.48	2.02	2.69
10	Pant Dhan-12	IP	22.0	12.5	247.7	306.9	10.21	3.74	2.15	3.33
11	Annada	IP	27.0	9.5	158.2	318.3	7.99	2.75	3.38	3.45
12	Dhala Heera	IP	17.9	8.5	200.1	312.2	6.62	2.72	2.70	3.13
13	Heera	IP	21.5	10.7	145.2	351.0	6.82	2.83	3.15	3.77
14	Kalyani-2	IP	21.4	11.3	156.3	240.5	7.32	3.08	2.92	3.67
15	Abhishek	IP	20.0	11.1	213.5	258.7	7.17	3.62	2.78	3.06
16	Pant Sugandh Dhan 15	IP	17.1	8.4	198.9	320.2	6.24	2.24	2.74	3.75
17	PR-116	IP	23.1	9.0	204.9	340.8	7.77	3.36	2.98	2.69
18	PR-120	IP	24.9	11.3	226.4	298.0	8.53	4.26	2.92	2.65
19	Ranbir Basmati-1	IP	26.6	12.6	207.8	255.2	8.17	4.19	3.26	3.01
20	Sattari	IP	19.7	12.8	117.7	242.1	6.77	3.01	2.91	4.25
21	Tara	IP	27.9	11.1	139.1	286.9	7.67	2.52	3.64	4.39
22	Vanaprava	IP	20.2	10.5	161.7	279.5	8.52	2.62	2.37	4.01
23	Hazari Dhan	IP	19.8	11.2	207.0	278.1	7.47	2.32	2.65	4.84
24	Sadabahar	IP	21.9	8.6	130.7	316.7	7.20	3.76	3.04	2.29
25	IR119	IP	18.7	10.4	208.3	261.3	6.70	2.89	2.79	3.59
26	Panindar	IP	26.3	13.4	175.9	288.7	7.64	3.31	3.44	4.04
27	VL Dhan-08	IP	26.1	15.1	214.8	234.6	9.44	4.43	2.76	3.40
28	IR36	IP	19.0	11.6	203.4	264.4	6.11	2.97	3.12	3.89
29	Ranbir Basmati-2	IP	26.3	18.3	152.2	265.9	8.15	3.57	3.23	5.12
30	Moroberkan	LR	27.0	17.5	204.2	263.6	8.44	4.26	3.20	4.10
31	AC 36722	LR	21.4	12.9	213.8	294.2	8.53	2.38	2.50	5.42
32	AC 36723	LR	20.6	11.0	172.2	265.1	9.27	2.63	2.23	4.17
33	AC 36732	LR	20.1	6.7	233.9	344.9	9.66	1.58	2.08	4.23
34	AC 36741	LR	25.3	10.5	173.6	276.4	7.76	2.22	3.26	4.73
35	AC 36726	LR	22.4	8.4	181.4	321.9	6.96	2.40	3.22	3.49
36	AC 36735	LR	19.1	16.3	232.3	252.3	6.73	2.69	2.85	6.06
37	AC 36754	LR	20.9	9.6	227.8	296.7	7.95	2.59	2.63	3.70
38	AC 36750	LR	28.5	10.5	179.7	284.9	8.96	3.23	3.18	3.24
39	AC 36734	LR	22.3	16.8	199.3	233.5	6.82	2.54	3.27	6.60
40	AC 36727	LR	21.9	12.9	142.5	297.5	7.36	3.50	2.98	3.69
41	AC 36762	LR	25.5	19.4	191.0	230.9	8.37	2.62	3.04	7.39
42	AC 35716	LR	26.5	11.1	121.8	292.2	7.62	2.42	3.47	4.60
43	AC 35717	LR	27.2	21.2	134.5	214.2	7.79	3.00	3.49	7.09
44	AC 35719	LR	23.8	10.4	217.9	289.6	10.16	3.16	2.35	3.30
45	AC 35729	LR	21.3	17.5	151.6	232.6	7.32	3.59	2.91	4.88
46	AC 35730	LR	26.3	12.3	155.6	273.4	5.79	2.87	4.54	4.28
47	AC 35731	LR	26.8	9.6	148.7	307.9	8.14	2.23	3.29	4.29
48	AC 36738	LR	19.4	16.8	122.9	219.6	8.24	3.38	2.35	4.97
49	AC 35736	LR	24.6	11.2	140.8	320.7	7.75	3.41	3.17	3.28
	CD 5%		0.88	0.88	11.01	9.30	0.30	0.19	0.13	0.32

Improved varieties (IP); landraces (LR); photosynthesis rate (An, µmol CO_2_ m^−2^ s^−1^); transpiration rate (Tr, mmol H_2_O m^−2^ s^−1^); water use efficiency (WUEi, µmol CO_2_ mmol^−1^ H_2_O); internal CO_2_ concentration (Ci, µmol CO_2_ mol^−1^); well-watered (WW); and drought stress (DS).

## Data Availability

Not applicable.

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
