# Peer review of "Unlocking the Nexus between Leaf-Level Water Use Efficiency and Root Traits Together with Gas Exchange Measurements in Rice (Oryza sativa L.)"

_plants, 2022, doi:10.3390/plants11091270_

Round 1

Reviewer 1 Report

The study was focused on investigation of the interplay between leaf-level water use efficiency and root traits together with gas exchange measurements in rice (Oryza sativa L.). The Authors demonstrated the relevance of water use efficiency with deeper rooting along with other root traits and gas exchange parameters. Forty-nine rice genotypes were evaluated in the basket method to examine the variation in leaf-level water use efficiency (WUEi) and its relation to root traits. Importantly, under drought stress conditions, landraces were more water-use efficient with superior root traits than improved genotypes.

The paper is quite interesting. However, I recommend some improvements:

  • Moderate English changes in the manuscript by the native speaker are required,
  • In Materials and Methods, a separate passage regarding statistical tests should be added,
  • The type of statistical test used should be included in figure captions, for example in Figures 4-5,
  • Table 1 contains a huge number of data, so it should be moved to a supplementary file, split into two smaller tables, or part of the data should be presented as a chart,
  • Some of the references should be replaced with the relevant newer citations.

Author Response

Mr. Kevin Wei

Section Manage Editor

Plants Editorial Office

Dear Mr. Wei:

We are pleased to submit a revised draft of our research article titled "Unlocking the nexus between leaf-level water use efficiency and root traits together with gas exchange measurements in rice (Oryza sativa L.) (Manuscript ID: plants-1708135)" to Plants Journal. We are grateful to both reviewers for their valuable suggestions and comments on the manuscript for sound improvement. We have incorporated all the changes suggested by the reviewers. We also appreciate the positive comments and suggestions. The changes were made in the manuscript using the track function. Responses to the reviewers' comments are provided below.

Reviewer 1

The study was focused on investigation of the interplay between leaf-level water use efficiency and root traits together with gas exchange measurements in rice (Oryza sativa L.). The Authors demonstrated the relevance of water use efficiency with deeper rooting along with other root traits and gas exchange parameters. Forty-nine rice genotypes were evaluated in the basket method to examine the variation in leaf-level water use efficiency (WUEi) and its relation to root traits. Importantly, under drought stress conditions, landraces were more water-use efficient with superior root traits than improved genotypes.The paper is quite interesting. However, I recommend some improvements:

  1. Moderate English changes in the manuscript by the native speaker are required,

Response: The manuscript was edited by a professional English editor in IRRI Communications.   

  1. In Materials and Methods, a separate passage regarding statistical tests should be added,

Response: As suggested, we have included a separate passage on statistical analysis to the Materials and Methods section in the revised manuscript (page 10).

  1. The type of statistical test used should be included in figure captions, for example, in Figures 4-5,

Response: The type of statistical test used has been included in the legends of Figures 3, 4, and 5 on pages 3-5 as per the reviewer's suggestion.

  1. Table 1 contains a huge number of data, so it should be moved to a supplementary file, split into two smaller tables, or part of the data should be presented as a chart,

Response: As suggested, Table 1 has been split into two tables in the revised manuscript and these have been renumbered.

  1. Some of the references should be replaced with the relevant newer citations.

Response: Older references were replaced by relevant newer citations in the revised manuscript.

Reviewer 2 Report

I completed the review of the article :Unlocking the nexus between leaf-level water use efficiency and root traits together with gas exchange measurements in rice (Oryza sativa L.).
It is an interesting and topical study in the context of the need to conserve water resources. Strong points are the high number of genotypes, the experimental design and the statistical tests. I have only a few specific comments:
L 95 and L96 please give examples of the investigated traits.
L 110 Figure 1 please specify what DAS stands for.
L 119 please specify what WW and DS are.
L 140 please include the units of measurement for all parameters.
L 172 according to your notations A should be An. I also don't see where Tr is on the figure.

Author Response

Reviewer 2

I completed the review of the article: Unlocking the nexus between leaf-level water use efficiency and root traits together with gas exchange measurements in rice (Oryza sativa L.). It is an interesting and topical study in the context of the need to conserve water resources. Strong points are the high number of genotypes, the experimental design and the statistical tests. I have only a few specific comments:

Response: Thanks for your valuable suggestions, which we have now incorporated in our corrections through the track function.  

  1. L 95 and L96 please give examples of the investigated traits.

Response: We have now incorporated examples of the investigated traits on page 3.

  1. L 110 Figure 1 please specify what DAS stands for.

Response: DAS stands for “days after stress.” This has been mentioned in the legend of Figure 1 on page 4 (line 149) of the revised manuscript.

  1. L 119 please specify what WW and DS are.

Response: The full forms of WW and DS are specified in line 156 of page 4 in the revised manuscript.

  1. L 140 please include the units of measurement for all parameters.

Response: Units of measurement of all parameters are included in lines 182 to 184 on page 6 and lines 191 to 193 on page 7 of the revised manuscript.

  1. L 172 according to your notations A should be An. I also don't see where Tr is on the figure.

Response: As per the suggestion, the A notation has been changed to An. Also, the notation E has been replaced by Tr in Figure 2 in the revised manuscript.
